# Antioxidant Activity of Crocodile Oil (*Crocodylus siamensis*) on Cognitive Function in Rats

**DOI:** 10.3390/foods12040791

**Published:** 2023-02-13

**Authors:** Krittika Srisuksai, Kongphop Parunyakul, Pitchaya Santativongchai, Narumon Phaonakrop, Sittiruk Roytrakul, Phitsanu Tulayakul, Wirasak Fungfuang

**Affiliations:** 1Department of Zoology, Faculty of Science, Kasetsart University, Bangkok 10900, Thailand; 2Bio—Veterinary Science (International Program), Faculty of Veterinary Medicine, Kasetsart University, Bangkok 10900, Thailand; 3Functional Ingredient and Food Innovation Research Group, National Center for Genetic Engineering and Biotechnology (BIOTEC), National Science and Technology Development Agency, Pathum Thani 12120, Thailand; 4Department of Veterinary Public Health, Faculty of Veterinary Medicine, Kasetsart University, Bangkok 10900, Thailand; 5Rresearch and Development Institute, Kasetsart University, Bangkok 10900, Thailand

**Keywords:** crocodile oil, antioxidant, cognitive function, hyperlipidemia, rat

## Abstract

Crocodile oil (CO) is rich in monounsaturated fatty acids and polyunsaturated fatty acids. The antioxidant activity and cognitive effect of monounsaturated fatty acids and polyunsaturated fatty acids have been largely reported. This work aimed to investigate the effect of CO on antioxidant activity and cognitive function in rats. Twenty-one rats were divided into three treatment groups: (1) sterile water (NS), (2) 1 mL/kg of CO (NC1), and (3) 3 mL/kg of CO (NC3). Rats underwent oral gavage once daily for 8 weeks. CO treatment decreased the triglycerides level significantly compared with that in the NS group. CO had a free radical scavenging ability greater than that of olive oil but had no effect on levels of antioxidant markers in the brain. Expression of unique proteins in the CO-treatment group were correlated with the detoxification of hydrogen peroxide. Rats in the NC1 group had better memory function than rats in the NC3 group. Expression of unique proteins in the NC1 group was correlated with memory function. However, CO did not cause a decline in cognitive function in rats. CO can be an alternative dietary oil because it has a hypolipidemia effect and antioxidant activity. In addition, CO did not cause a negative effect on cognitive function.

## 1. Introduction

Crocodiles have high economic and medicinal values. *Crocodylus siamensis* (Siamese crocodile) is an endangered species of freshwater crocodile, which was originally distributed throughout Southeast Asia [1]. Currently, crocodiles can be farmed by providing a suitable habitat because their skin and meat are in demand for a niche market [2]. Therefore, waste is generated from the crocodile product industry (e.g., internal organs, fat). Crocodile fat can be extracted to make crocodile oil (CO), which has been used for the treatment of ailments ranging from skin conditions to cancer [3,4]. The current usage practices of CO are used as ointments for skin conditions due to its possibility to accelerate wound healing and reduce scar regeneration by downregulation of the expression of the p38 mitogen-activated protein kinase signaling pathway [5,6]. Moreover, CO exhibits both antimicrobial and anti-inflammatory activities [3]. CO could reduce DNA damage and increase the cell-cycle regulators, which modulate the inflammation [7]. Interestingly, oral supplementation of CO does not cause acute toxicity in Wistar rats [4]. Previously, our study found CO supplementation could preserve hepatic mitochondrial structure and increase energy metabolic activity [8]. The other study indicated that CO increases hepatic detoxifications by altering CYP1A2 expression in high fat diet rats [9]. However, scientific evidence for the antioxidant activity and cognitive function of CO is lacking.

Oxygen is indispensable for life. Cells use oxygen to generate adenosine triphosphate, which leads to the creation of free radicals by mitochondria [10]. A ‘free radical’ is an atom or molecule that has an unpaired electron and is, therefore, unstable [11]. The unstable radical tends to become stable through electron pairing with biological macromolecules such as proteins, lipids, and DNA, thereby causing damage to proteins and DNA [12,13]. In general, cells have an enzymatic and non-enzymatic antioxidant defense system to protect against such damage [14,15]. Antioxidants delay or inhibit the oxidation of lipids or other molecules by stopping the initiation or propagation of oxidizing chain reactions [16]. However, if the production of free radicals exceeds the ability of the cell to defend against these substances, then oxidative stress results [17].

The brain is especially sensitive to oxidative stress owing to its high rate of oxygen consumption, abundant lipid content, and low levels of antioxidant enzymes when compared with other tissues [18]. Numerous experimental and clinical studies have demonstrated that oxidative stress plays a key role in the loss of neurons and progression to cognitive impairment [10,19]. Increasing the intake of antioxidants in the human diet is an important way to minimize oxidative damage [20]. Antioxidants can be found in various food products such as vegetables and oils, the consumption of which can reduce the risk of chronic diseases [14]. Recently, synthetic antioxidants such as butylated hydroxy toluene (BHT), butylated hydroxy anisole (BHA), and propyl gallate (PG) have been used extensively, but they have been associated with toxicity and some side effects (e.g., carcinogenesis) [15,21]. Thus, the number of studies on ‘natural antioxidants’ has increased [22].

CO is rich in monounsaturated fatty acids (MUFAs) and polyunsaturated fatty acids (PUFAs). The oleic acid (OA) has the highest fatty acid content in CO [5,23,24]. OA has been reported to have antioxidant activity thanks to inhibition of oxidative stress-induced toxicity in *Caenorhabditis elegans* [25]. OA consumption has been shown to increase the glutathione (GSH) level in streptozotocin-induced diabetic rats fed a high-cholesterol diet [26]. In addition, OA intake can protect against cognitive decline during aging and improve cognitive function [27]. The second highest fatty acid content in CO is linoleic acid (LA) [5,23,24]. Ly and colleagues revealed that LA isolated from the oil of *Sambucus williamsii* seeds had antioxidant activity [28]. LA activates antioxidation through the adenosine monophosphate-activated protein kinase autophagic signaling pathway [29]. LA, the main source of oxidized linoleic acid metabolites, could regulate neuronal morphology and neurotransmission [30]. Proteomics analysis is a powerful tool used widely to investigate protein expression and understand complex physiological processes at the protein level [31]. There are many previous reports that used proteomic analysis to substantiate the results and find the association between diet, oxidative damage, and cognitive function [32,33].

We hypothesized that the antioxidant activity of CO could maintain cognitive function. As per our knowledge, our finding is the first report of the antioxidant activity of CO and the effect of CO on brain protein expression with cellular oxidative and cognitive managements. In this study, we aimed to clarify the effect of CO on antioxidant activity and cognitive function in rats.

## 2. Materials and Methods

### 2.1. Materials

CO was extracted using the wet cold-pressed method as described by Santativongchai et al. [23]. Abdominal fat of *C. siamensis* was obtained as a by-product from a crocodile farm in Nakhon Pathom, Thailand. The fat samples were pressed through two layers of filter cloth with distilled water at the proportion of 1:1 (*w*/*v*). The solution was left until the separation of the mixture was observed and the clear oil fraction was collected, evaporated, and stored at 25 °C until used. Palm oil (PO) and extra virgin olive oil (EVOO) were purchased from a local supermarket in Bangkok (Thailand). 2,2-diphenyl-1-picrylhydrazyl (DPPH) was obtained from MilliporeSigma (Burlington, MA, USA). GSH standard was purchased from Merck (Darmstadt, Germany). Ethyl acetate and ethyl alcohol were sourced from Thermo Fisher Scientific (Waltham, MA, USA). All other reagents and chemicals used were of analytical grade.

### 2.2. Fatty Acid Composition

Identification and quantification of the fatty acids were carried out by gas chromatography (Agilent 7890B, Santa Clara, CA, USA) according to the method of Buthelezi et al. [3] with slight modifications. The instrument was filled with a flame ionization detector and fatty acid separation was carried out on a fatty acid methyl ester column (CP-Sil 88, Agilent, Santa Clara, CA, USA) with a length of 100 m, an internal diameter of 0.25 mm, and a stationary phase film thickness of 0.20 μm. Identification of fatty acids was achieved by comparison of the retention times with authentic standard fatty acid methyl esters.

### 2.3. Free Radicals Scavenging Activity of CO In Vitro

The DPPH assay was carried out based on the method described by Akmal and Roy [14] with slight modifications. If a hydrogen atom or electron is transferred to the DPPH radical (DPPH**·**), then the absorbance at 517 nm decreases proportionally to the increase in the number of non-radical forms of DPPH. Briefly, 50 µL of each concentration of CO, PO, EVOO, and BHT was added to 150 µL of DPPH in ethyl acetate (150 µM). After vortex mixing, the mixture was incubated for 30 min at room temperature and the absorbance measured at 517 nm. BHT was used as the positive control. All samples were tested in triplicate. The differences in absorbance of each oil, BHT, and control (DPPH alone) was documented. The radical scavenging activity was calculated as % inhibition:% inhibition = [(control − test)/control] × 100(1)

The half-maximal inhibitory concentration (IC_50_) was determined as the concentration of oil that elicited a 50% decrease in the absorbance against control.

### 2.4. Animals

The protocol of animals was approved by the Animal Use and Care Committee of Kasetsart University Research and Development Institute, Kasetsart University, Thailand (ACKU61-VET-088) according to the guidelines for animal care and use under the Ethical Review Board of the office of National Research Council of Thailand (NRCT). Twenty-one male Wistar rats (7 weeks) were obtained from Nomura Siam International (Samutprakarn, Thailand). Rats were allowed to acclimatize to the laboratory environment for 1 week before experimentation. Rats were kept under controlled conditions with a 12 h light–dark cycle at room temperature (25 ± 2 °C) and relative humidity (60–70%). They had free access to water and rat chow (Nomura Siam International).

### 2.5. Experimental Design

According to Naphatthalung et al. [34], the animals were treated with CO at dose 1 or 3 mL/kg bodyweight. Hence, healthy rats were divided randomly into three groups of seven: (1) rats fed sterile water (NS group); (2) rats fed CO (1 mL/kg bodyweight) (NC1 group); and (3) rats fed CO (3 mL/kg bodyweight) (NC3 group). Rats in the same group were housed together. All groups had free access to water and rat chow (51% carbohydrate, 4.6% fat, and 24.90% protein: Nomura Siam International) until the end of the experiment. Sterile water and CO were administered via the gavage once daily for 8 weeks.

### 2.6. Food Intake, Energy Intake, Bodyweight Gain, and Bodyweight of Rats

The daily food intake of each rat was measured by weighing the remaining rat chow; food spillage was accounted for when measuring food intake. Daily energy intake per rat was calculated as described by Gong et al., 2016 [35]:daily energy intake = (food intake × total energy of the chow diet) + energy from the treatment(2)

Total energy of the rat diets was 3.040 kcal/g, and energy of CO was 12 kcal/1.5 mL. The body weight of all rats was monitored once a week throughout the experiment.

### 2.7. Morris Water Maze (MWM) Test

After 8 weeks, the MWM test was used to measure spatial learning and memory. The MWM test was a modified form of the method described by Vorhees and Williams [36]. The MWM test was undertaken in a water pool (diameter = 200 cm) and was filled to a depth of 35 cm with clear water maintained at 26 ± 1 °C. The water pool was located in a large and quiet test room surrounded by several different cues, which were visible from the water pool and could be used by rats for spatial orientation. The position of the cues remained unchanged throughout the study. Testing was carried out over 6 days, the first day was regarded as screening for locomotor activity. Testing consisted of an acquisition phase and a probe phase. The acquisition phase was undertaken for 4 consecutive days of training with 4 trials per day. A transparent platform (diameter = 15 cm) was submersed 2 cm beneath the water surface. The platform was located in a designated target quadrant. Rats had 120 s to locate the hidden platform. If a rat did not find the platform within 120 s, then it was guided to the platform. The time taken to reach the platform was recorded. The probe test was performed 24 h after the last trial of the acquisition phase. In the probe test, the platform was removed and each rat was allowed to swim for 60 s. The swimming activity of each rat was tracked via a video camera positioned directly above the center of the pool. Latency time to find the platform, the time spent in the target quadrant, and average swimming speed were analyzed using a computerized video tracking system (SMART 3.0.04, Panlab, Barcelona, Spain).

### 2.8. Sample Collection

Rats were killed with pentobarbital (60 mg/kg) after fasting for ≥6 h. Blood was collected from the left ventricle for measurement of lipid profiles. Whole blood was centrifuged at 2000× *g* for 10 min at 4 °C. Serum was collected and stored at –20 °C. Lipid profiles (cholesterol, triglyceride, high-density lipoprotein-cholesterol (HDL-C), low-density lipoprotein-cholesterol (LDL-C)) levels were determined using an automatic analyzer (7080 series; Hitachi, Tokyo, Japan).

After collecting blood, rats were decapitated immediately. Whole-brain samples were rinsed in ice-cold physiologic (0.9%) saline to remove extraneous materials. Brains in each group were homogenized in ice-cold phosphate buffer and divided into two parts. One part was subjected to an analysis of antioxidant markers. The other part was diluted with acetone at a 2:1 (*v*/*v*) ratio and then centrifuged at 10,000× *g* for 15 min. Supernatants were stored at −80 °C for proteomics analysis. Protein concentrations were determined by the Lowry assay using bovine serum albumin as a standard protein [37].

### 2.9. Antioxidant Markers

#### 2.9.1. GSH

GSH activity was determined according to the method described by Lacoste et al. [38] with some modifications. Both 1 mL of homogenate and 1 mL of 10% trichloroacetic acid were centrifuged at 8960× *g* for 10 min. Then, 42 µL of supernatant was mixed with 168 µL of 5,5′-dithiobis (2-nitrobenzoic acid) and 42 µL of phosphate buffer in a final volume of 252 µL in a microplate. The absorbance of the supernatant was read at 412 nm. A standard curve was established using GSH (0, 0.1, 0.2, 0.4, 0.6, 0.8, 1 mM). Results are expressed as the mM of GSH/mg of protein.

#### 2.9.2. Glutathione-S-Transferase (GST)

The GST activity was estimated according to the method of Ozcelebi et al. [39] with slight modifications. The reaction mixture consisted of 1-Chloro-2,4-dinitrobenzene (CDNB; 1 mM), GSH (1 mM), phosphate buffer (0.1 M, pH 6.5), and brain homogenate. Changes in the absorbance were recorded at 340 nm and enzyme activity was calculated as nmol CDNB conjugate formed/min/mg of protein.

#### 2.9.3. Catalase (CAT)

CAT activity was assayed according to the method described by Mokhtari-Zaer et al. [40] with some modifications. This method is based on the ability of CAT to decompose hydrogen peroxide (H_2_O_2_) measured at 240 nm. Briefly, the assay mixture consisted of 1.95 mL of potassium phosphate buffer (0.05 M, pH 7.0), 1 mL of H_2_O_2_ (10 mM), and 0.05 mL of brain supernatant in a final 3 mL volume. The reaction was started by H_2_O_2_ addition and the reduction in absorption was measured by continuous monitoring at 240 nm for 3 min. CAT activity was expressed as units/mg of protein.

### 2.10. Proteomic Analysis

#### 2.10.1. Sample Preparation

In-gel digestion was determined according to the method of Losuwannarak et al. [41]. To reduce the disulfide bonds of total protein isolated from rat brains, dithiothreitol (10 mM) in ammonium bicarbonate (10 mM) was added. Reformation of disulfide bonds in proteins was blocked by alkylation with iodoacetamide (30 mM) in ammonium bicarbonate (10 mM). Protein samples were digested with sequencing-grade porcine trypsin (ratio of 1:20; Promega, Mannheim, Germany) and incubated overnight at 37 °C. Tryptic peptides were dried using a speed vacuum concentrator (Thermo Fisher Scientific, Waltham, MA, USA) and resuspended in 0.1% formic acid for nano-liquid chromatography-tandem mass spectrometry (nanoLC-MS/MS).

#### 2.10.2. LC-MS/MS

Samples of tryptic peptides were prepared for injection into a nano/capillary LC System (Ultimate 3000; Thermo Fisher Scientific, Waltham, MA, USA) coupled to a LC-MS system (HCTUltra; Bruker Daltonics, Hamburg, Germany) equipped with a nano-captive spray ion source. Briefly, 5 µL of peptide digests were enriched on a µ-pre-column (300 µm i.d. × 5 mm C18 Pepmap 100, 5 µm, 100 A Thermo Scientific) separated on a column (75 μm I.D. × 15 cm) and packed with an Acclaim PepMap RSLC C18 column (2 μm, 100Å, nanoViper; Thermo Scientific). The C18 column was enclosed in a thermostatted column oven set to 60 °C. Solvent A and B (containing 0.1% formic acid in water and 0.1 % formic acid in 80% acetonitrile, respectively) were supplied on the analytical column. A gradient of 5–55% solvent B was used to elute peptides at a constant flow rate of 0.30 μL/min for 30 min. Electrospray ionization was carried out at 1.6 kV using the nano-captive spray ion source. Nitrogen was used as a drying gas (flow rate ~50 L/h). Collision-induced-dissociation product ion mass spectra were obtained using nitrogen gas as the collision gas. MS and MS/MS spectra were obtained in positive-ion mode at 2 Hz over the range *m/z* = 150–2200. The collision energy was adjusted to 10 eV as a function of the *m/z* value. The LC-MS of each sample was performed in triplicate.

A differential analysis software (DeCyderMS; GE Healthcare, Chicago, IL, USA) was used to quantify the proteins in individual samples. The Mascot search engine was used to correlate the MS/MS spectra to a Macaca protein database maintained by UniProt [42,43]. Searches were undertaken with a maximum of three missed cleavages, carbamidomethylation of Cys as a fixed modification, and oxidation of Met as variable modifications. The protein levels in each sample were expressed as log2 value.

### 2.11. Data and Statistical Analysis

A Venn diagram was used to show the differences between the protein lists originating from differential analyses. Data were searched against the UniProt database for protein identification to understand the molecular function and biological processes. The STITCH database (version 5) was used to predict networks for chemical-protein and protein–protein interactions.

Statistical analyses were undertaken using GraphPad Prism version 9.5 (San Dieago, CA, USA). Data are the mean ± SEM. Escape latency on the MWM test was analyzed by mixed model one-way analysis of variance (ANOVA) following Tukey’s post hoc test. Other data were analyzed by one-way ANOVA followed by Tukey’s post hoc test. *p* < 0.05 was considered significant.

## 3. Results

### 3.1. Fatty Acid Profiles of CO

The fatty acid composition of CO is formed by a mixture of saturated fatty acids (SFAs) and unsaturated fatty acids (UFAs) classified according to the number of unsaturated bonds as MUFAs or PUFAs (Table 1). CO contains high levels of UFAs (69.34%) and low levels of SFAs (26.40%). It was shown that OA (C18:1n9) (41.07%), LA (C18:2 n6c) (21.08%), and palmitic acid (PA)(C16:0) (19.92%) were found to be highest content in CO.

### 3.2. In Vitro Antioxidant Activity of CO Using the DPPH Assay

The free radical scavenging activity of CO, PO, EVOO, and standard BHT is presented in Figure 1. CO, PO, and EVOO showed a dose-dependent inhibitory effect on the ability to scavenge DPPH**·** free radicals (Figure 1a). The IC_50_ (in mg/mL) of CO, PO, EVOO, and BHT (standard) was 85.77 ± 1.81, 64.66 ± 1.09, 115.71 ± 4.79, and 5.59 ± 0.04, respectively (Figure 1b). The ability of different oils and BHT to scavenge DPPH**·** free radicals was in the order BHT > PO > CO > EVOO.

### 3.3. Effect of CO on Food Intake, Energy Intake, Body Weight Gain, and Body Weight

The food intake, energy intake, body weight gain, and body weight of rats after 8 weeks of treatment are shown in Table 2 and Figure 2. Rats in the NC1 group and NC3 group had significantly decreased food intake compared with that in the NS group. However, there was no significant difference in energy intake, body weight gain, or body weight.

### 3.4. Effect of CO on Serum Lipid Profiles

After 8 weeks of treatment, the triglycerides level decreased significantly in the NC1 group and NC3 group compared with that in the NS group. However, there was no significant change in the level of cholesterol, HDL-C, or LDL-C among groups (Table 3).

### 3.5. Effect of CO on Spatial Learning and Memory

Spatial learning and memory were measured using the MWM test. The latency time to the platform in the NC1 group and NC3 group was not significantly different compared with that in the NS group (Figure 3a). However, the time spent in the target quadrant was increased significantly in the NC1 group, compared with that in the NC3 group, but was not significantly different compared with that in the NS group (Figure 3b). There was no significant difference between groups for the average swimming speed (Figure 3c).

### 3.6. Effect of CO on Markers of Antioxidant Activity in the Brain

The antioxidant activity of GSH, GST, and CAT are shown in Figure 4. There was no significant difference between groups in levels of GSH, GST, or CAT.

### 3.7. Effect of CO on Expression of Antioxidant-Related Proteins in the Brain

A total of 4969 proteins were identified by “shotgun” proteomics analysis. Of these, 60 were present in all three groups. The Venn diagram in Figure 5a shows the number of differentially expressed proteins among groups. There were 106, 269, and 134 unique proteins expressed in the NS, NC1, and NC3 groups, respectively.

To identify the effect of CO on expression of protein related to antioxidant activity in the brain, unique proteins expressed in NC1 group, NC3 group, and shared proteins expressed in NC1 and NC3 groups were identified using the UniProt database. Six proteins were associated with antioxidant activity (Table 4). The proteins and chemicals that interacted with these proteins were analyzed by STITCH (Figure 5b). Protein interactions demonstrated that these proteins were strongly related to peroxidase activity and oxidative phosphorylation, which have major roles in antioxidant activity.

### 3.8. Effect of CO on Expression of Cognition-Related Proteins in the Brain

Rats in the NC1 group had significantly increased memory compared with that for rats in the NC3 group (Figure 3b). To better understand how the NC1 group had improved memory, unique proteins in the NC1 group involved with cognitive function were identified using the UniProt database (Table 5). The proteins and chemicals that interacted with unique proteins involved in cognitive function were analyzed by STITCH (Figure 6). We demonstrated that unique proteins were strongly related to the glutamate receptor signaling pathway and neurotransmitter transport.

## 4. Discussion

The brain is particularly vulnerable to production of free radicals because of its high oxygen consumption rate, abundant lipid content, and limited amount of antioxidant capacity [18]. Dietary fatty acids have important roles in human health because they influence physiological functions: the physical state of the cell membrane, hormone binding, signal transduction, and eicosanoid production [44]. CO has been found to be rich in MUFAs and PUFAs (Table 1). Similar to previous studies, the OA, LA, and PA were found to be highest composition in CO [5,23,24]. CO has levels of OA and LA that are four-times higher than those in fish oil [45]. However, the three fatty acids with the highest content of CO from *C. niloticus* were OA (20%), PA (15%), and LA (4%) [3]. This suggested that the fatty acid composition of CO from *C. siamensis* has more UFAs than CO from *C. niloticus*. Numerous studies have reported that MUFAs and PUFAs have antioxidant activity and neuroprotective effects in clinical and animal studies [26,28,29,46,47].

The ability of CO to act as a hydrogen/electron donor or scavenger of free radicals was determined by an in vitro antioxidant assay using DPPH. The positive control was BHT. Measurement of the ability to scavenge DPPH**·** free radicals involves use of colorimetric methods that are simple and reproducible [48]. If the DPPH**·** free radical is scavenged, the reaction mixture undergoes discoloration with decreasing absorption that indicates formation of a stable diamagnetic molecule [49]. The ability to scavenge free radicals by CO, PO, and EVOO was observed at a concentration of 10–200 mg/mL. All of the oils showed an ability to scavenge DPPH**·** free radicals that increased in a concentration-dependent manner (Figure 1a). However, the free radical scavenging ability of CO, PO, and EVOO in the DPPH assay was lower than that of the standard antioxidants (BHT) (Figure 1b). In general, vegetable oils are good for human health because they have high levels of UFAs and other phytochemical compounds [50]. We found that PO had the highest ability to scavenge free radicals compared with that of CO and EVOO (Figure 1). PO has a high concentration of lipophilic antioxidants such as tocopherols and tocotrienols [51]. In addition, synthetic antioxidants (e.g., BHT, BHA, PG) are added in the PO industry to retard oxidation reactions [22]. However, we demonstrated that the free radical scavenging ability of CO was greater than that of EVOO (Figure 1). Previous study found that CO mixed with herbal oil (Phlai, Tumeric, and Black galingale) serves as an effective free radical scavenger [52]. The free radical scavenging of CO could be related to the tocopherols, which are powerful scavengers [53]. However, this is the first study to report the free radical scavenging of CO. The dietary intake of antioxidants is imperative to protect cells from damage caused by free radicals [12].

We demonstrated that the food intake of the NC1 group and NC3 group was less than that of the NS group by 1–3 g, which indicated the adjustment of energy intake [54]. In accordance with previous findings, dietary fatty acids did not affect energy intake among any groups [55]. The effect of dietary fat on bodyweight is dependent upon the amount of fats used in the diet and the fatty-acid composition [56]. Saturated fatty acids have been shown to produce higher levels of weight gain compared with other types of fatty acids [55]. Moreover, SFA-fed rats can develop insulin resistance in adipocyte tissue [44]. Insulin induces fatty-acid synthesis via acetyl-CoA carboxylase (ACC) activity in the liver [57]. However, UFAs such as MUFAs and PUFAs do not induce weight gain in humans or rodents [58,59,60]. Yang et al. [61] found that UFA-rich diets could prevent weight gain by increasing levels of an enzyme that activates the β-oxidation of fatty acids and lowers the insulin concentration in hamsters. Bodyweight gain and bodyweight did not change among groups (Table 2 and Figure 2).

Several studies have shown that the composition of dietary oils can affect lipid profiles in blood [62,63]. The NC1 group and NC3 group had a significantly decreased triglycerides level compared with that in the NS group (Table 3). The plasma triglycerides level is affected by absorption of dietary fat in the intestine, very low-density lipoprotein (VLDL) synthesis in the liver, as well as blood clearance of chylomicron and VLDL [64]. Consistent with our previous study, the findings indicated that CO administration significantly ameliorated hepatic fat accumulation by greater decrease in the total surface area of lipid droplets when compared to SFA-rich oil [8]. Researchers have found that PUFAs reduce the hepatic synthesis of fatty acids, which decreases the triglyceride levels in the liver [65]. Qi et al. [64] showed that PUFAs increased lipoprotein lipase (LPL) activity in mice fed a PUFA-rich diet for 4 weeks. LPL (which hydrolyzes triglycerides in lipoproteins and promotes cellular uptake of free fatty acids) plays an important part in maintaining the triglycerides concentration in blood [66]. Furthermore, CO contains a large fraction of OA, and triglyceride content in the bloodstream has been reported to decrease after consumption of a diet rich in OA [62,67]. Our current results suggest that CO could play a beneficial effect on lipid profiles by improving triglyceride levels and maintaining HDL-C levels after 8 weeks of experiments. Reducing the plasma level of triglycerides is important because it can assist in the development of new or improved pharmacological approaches to treating hypertriglyceridemia [68].

The MWM test was used to explore the learning and memory abilities of rats. The acquisition phase was used to test the ability of learning and short-term memory. The probe phase was employed to test long-term memory [69]. The NC1 group and NC3 group did not show a significant difference on MWM test compared with that in the NS group (Figure 3), which suggests that CO did not cause a decline on learning and memory function. Similar to our study, García-Cerro and colleagues showed that OA-treated mice did not change their performance in the MWM test compared with control mice [59]. Moreover, OA and LA intake could maintain learning and memory functions in the elderly people [27,70].

Oxidative stress and the antioxidant system have important roles in the development of degenerative diseases [71,72]. Oxidative stress damages cellular components and results in alterations in membrane properties, such as fluidity, ion transport, enzyme activities, and protein crosslinking [72]. GSH, GST, and CAT are common biomarkers of antioxidant activity. Dietary consumption of CO did not affect the level of GSH, GST, or CAT in the rat brain (Figure 4). Olive oil (which is rich in OA) has no effect on CAT activity in the rat brain [73]. For rats receiving an olive oil-enriched diet started from pregnancy until the weaning of pups, GSH in the hippocampus showed a similar level to that in the hippocampus of control rats [74]. Hence, CO did not appear to have negative effects on antioxidant levels in the rat brain in our study.

The results showed that CO has free radical scavenging abilities (Figure 1). To better understand the antioxidant effect of CO, unique proteins related to antioxidant activity in CO treatment were evaluated. We found that six proteins (peroxiredoxin 2 (Prdx2), CAT, haemoglobin subunit alpha-1/2 (Hba1), NADH-ubiquinone oxidoreductase core subunit 1 (Ndufs1), riboflavin kinase (Rfk), peroxidase (Pxdn)) were expressed specifically in the CO-treatment group. Prdx2 is an important antioxidant enzyme in the central nervous system [75]. Reports have shown that Prdx2 can reduce reactive oxygen species (ROS) production by catalyzing H_2_O_2_ to water, which maintains intracellular redox homeostasis and attenuates various neurological disorders [76,77]. CAT has one of the highest turnover rates among all biological enzymes; it detoxifies H_2_O_2_ by reducing this molecule to molecular oxygen (O_2_) and H_2_O [78]. Hemoglobin (Hb) is assembled into heterotetramers composed of two α-globin and two β-globin polypeptides [79]. The α- and β-chains of Hb are expressed in the mammalian brain [80]. Hb contains a heme prosthetic group, which binds not only O_2_ and carbon dioxide but also to nitric oxide, thereby allowing it to protect cells against oxidative and nitrosative stresses [81]. In mesangial cells of the kidney, Hb has been shown to act as antioxidative defense by scavenging superoxide anions and H_2_O_2_ [82]. Ndufs1 is one of the core subunits of mitochondrial complex I, which regulates mitochondrial oxidative phosphorylation and ROS production [83]. Knockdown of Ndufs1 expression impairs mitochondrial O_2_ consumption and increases ROS production in neurons and cardiomyocytes [83,84]. Rfk is required for the conversion of riboflavin into flavin mononucleotide, which is adenylated by FAD synthetase to generate flavin adenine dinucleotide (FAD) [85]. FAD is a coenzyme for glutathione reductase, which mediates GSH regeneration [86]. Pxdn are a large family of enzymes that catalyze the oxidation of a wide variety of organic and inorganic compounds using H_2_O_2_ as an electron-acceptor [87]. These peroxidases include GPx, myeloperosidase, eosinophil peroxidase, uterine peroxidase, lactoperoxidase, salivary peroxidase, and thyroid peroxidase [88]. Antioxidants affect almost all unique proteins involved in H_2_O_2_ detoxification. H_2_O_2_ represents a key target intervention because it is neither a radical nor an ion, thus it can cross the cell membrane and affect cellular structures distant from its origin [89]. H_2_O_2_ is one of the main ROS leading to oxidative stress, DNA damage, and cell death [90]. Therefore, we suggested that CO has antioxidant activity that is associated with H_2_O_2_ detoxification.

Rats in the NC1 group had better memory function than rats in the NC3 group (Figure 3b). To better understand how treatment in the NC1 group could improve memory, the unique proteins in the NC1 group involved with cognitive function were evaluated. Three proteins (glutamate receptor ionotropic delta 2 (Grid2), liprin-α3 (also known as Ppfia3), breast carcinoma amplified sequence 1 (Bcas1)) were expressed specifically in the NC1 group. Impairment of *Grid2* in mice has been shown to lead to poor motor learning and memory dysfunction [91]. Grid2 is a protein that belongs to the delta receptor subtype of the ionotropic glutamate receptor, which includes N-methyl-D-aspartate (NMDA) and non-NMDA receptors. Increased numbers of NMDA receptors lead to improved learning capacity, mediate synaptic plasticity, and are critical for the encoding of experiences and memories [92,93,94]. Proteins of the liprin-α family are core synaptic scaffolds and important for synapse maturation [95]. Liprin-α3 is expressed mainly in presynaptic and postsynaptic regions in the hippocampus, which are associated with cognitive function. Liprin-α3 is involved in presynaptic plasticity and release of synaptic vesicles, particularly in excitatory synapses [96]. Dysfunction or depletion of liprin-α3 leads to impairments in the docking and exocytosis of synaptic vesicles, which have a presynaptic role in synaptic transmission [97]. Bcas1 is expressed in oligodendrocytes and Schwan cells. Depletion of Bcas1 has been shown to lead to hypomyelination in mice [98]. Myelination contributes to memory consolidation by increasing functional coupling between the neuronal ensembles encoding experience [99]. We found that a high dose of CO reduced the memory function compared with that using a low dose of CO. CO is rich in MUFAs and PUFAs, but it also contained SFAs (Table 1). PA, highest SFAs in CO, could induce brain oxidative stress and lead to impairment in learning and memory in rodents [100,101,102]. However, a high dose of CO did not significantly differ from that in the NS group. Previous study indicated that antioxidant compounds could preserve and/or ameliorate cognitive impairment via free radical suppression [103]. Therefore, antioxidant activity of CO could be effective in preserving cognitive function during high-dose diets of fat. However, the molecular mechanism underlying the antioxidant activity of CO on cognitive function requires further study.

## 5. Conclusions

CO had free radical scavenging ability greater than that of EVOO and had antioxidant activity through H_2_O_2_ detoxification. Moreover, CO could reduce the triglyceride level. Interestingly, CO supplementation did not cause a decline in cognitive function. It is notable that this is the first study of the antioxidant activity of CO and the protective effect of CO on brain function in rats. Therefore, CO could be the alternative supplementary oil to treat oxidative stress as well as hypertriglyceridemia in the future.

## Figures and Tables

**Figure 1 foods-12-00791-f001:**
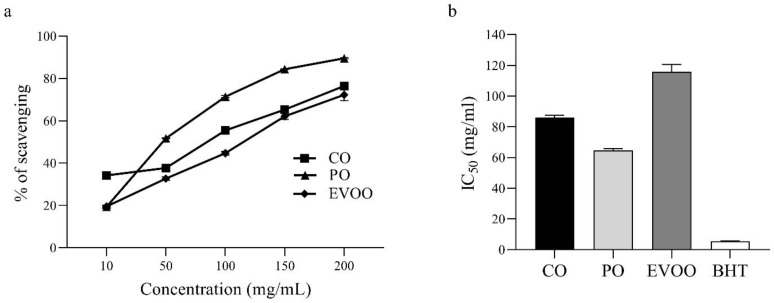
Antioxidant activity of oils determined using the DPPH assay. (**a**) Free radical scavenging ability. (**b**) IC_50_ of CO, PO, EVOO, and BHT. Values are the mean ± SEM.

**Figure 2 foods-12-00791-f002:**
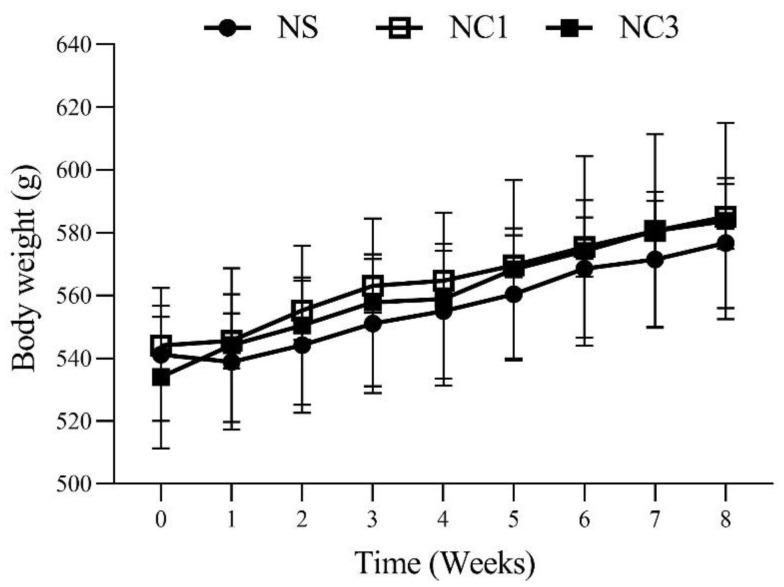
Bodyweight in rats on the 8 week treatment. Values are the mean ± SEM.

**Figure 3 foods-12-00791-f003:**
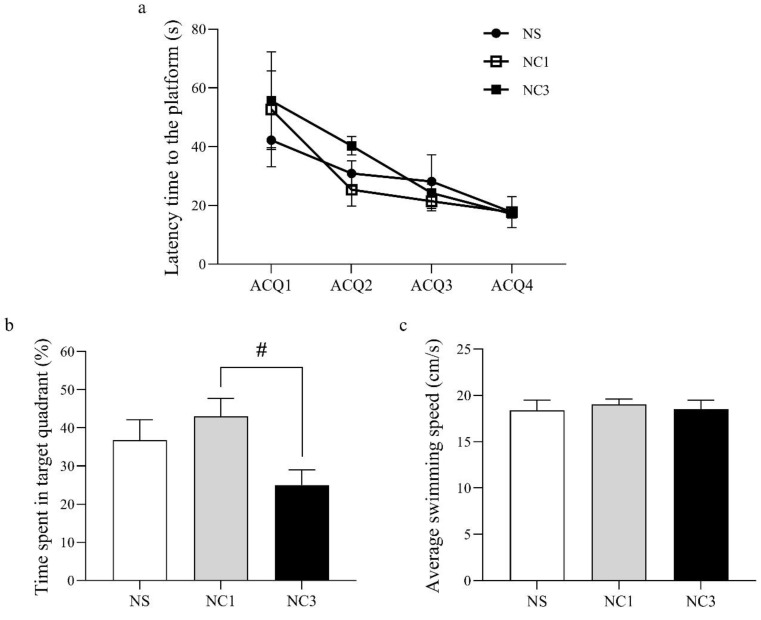
The learning and memory abilities in different groups of rats as assessed by the Morris water maze test. (**a**) Latency time to the platform. (**b**) Percentage of time spent in the target quadrant. (**c**) Average swimming speed. Values are the mean ± SEM. ^#^ *p* < 0.05 compared with the NC1 group.

**Figure 4 foods-12-00791-f004:**
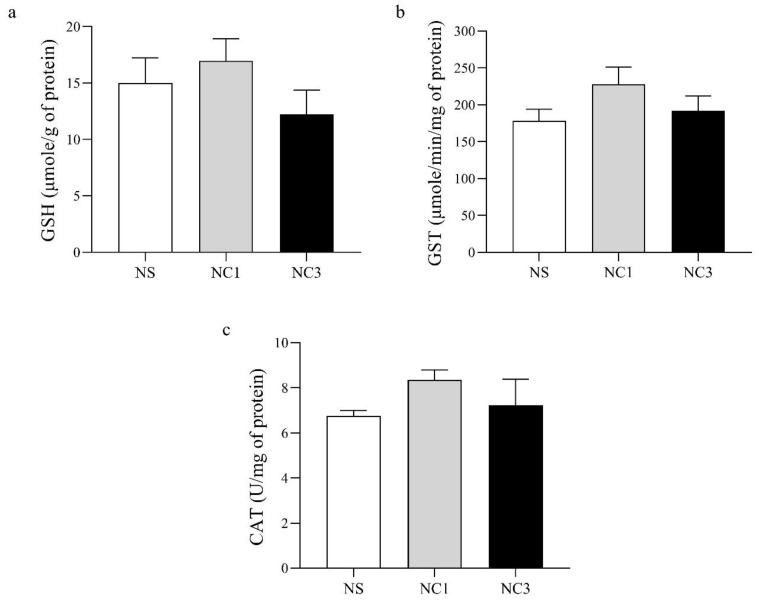
Effects of crocodile oil on markers of antioxidant activity. (**a**) Glutathione (GSH) level. (**b**) Glutathione-S-transferase (GST) level. (**c**) Catalase (CAT) level. Values are the mean ± SEM.

**Figure 5 foods-12-00791-f005:**
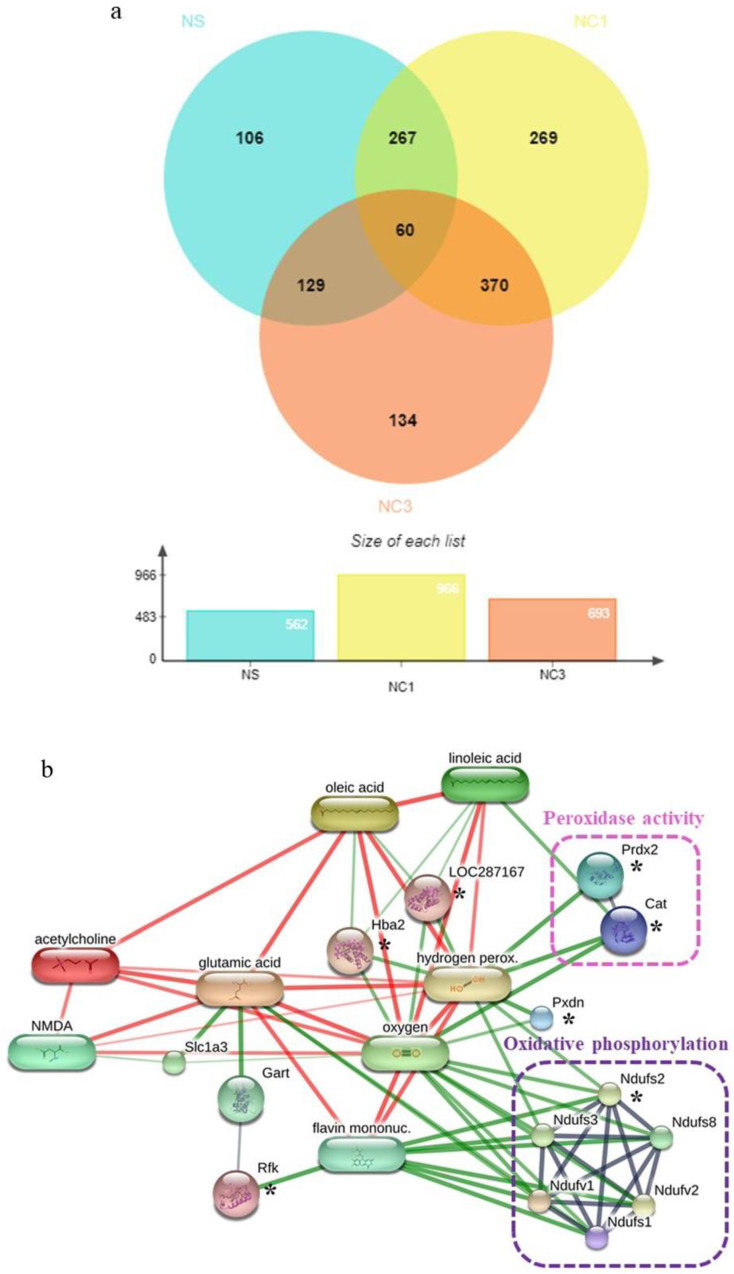
Comparative analyses and analyses of interaction networks proteins expressed in the brain. (**a**) Venn diagram of unique proteins and shared proteins among the three groups. NS, rats fed with sterile water; NC1, rats fed with CO (1 mL/kg bodyweight); NC3, rats fed with CO (3 mL/kg bodyweight). (**b**) Networks of chemical–protein and protein–protein interactions of unique proteins expressed in NC1 group, NC3 group, and shared proteins expressed in NC1 and NC3 groups as well as the pathways involved in antioxidant activity. * indicate the unique proteins expressed in NC1 group, NC3 group, and shared proteins expressed in NC1 and NC3 groups which related to antioxidant activity.

**Figure 6 foods-12-00791-f006:**
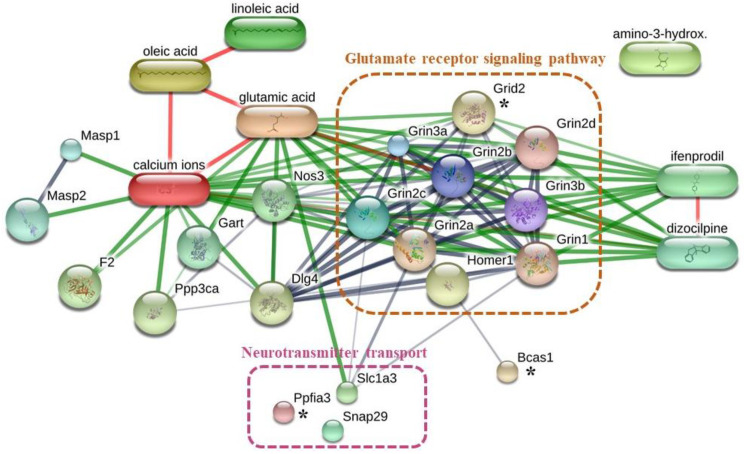
Networks of chemical–protein and protein–protein interactions of unique proteins expressed in NC1 group and pathways involved in cognitive function. * indicate the unique proteins expressed in NC1 group which related to cognitive function.

**Table 1 foods-12-00791-t001:** Fatty acid composition of crocodile oil analyses by gas chromatography.

Fatty Acid	Composition (%)
Lauric acid (C12:0)	0.11 ± 0.036
Myristic acid (C14:0)	0.57 ± 0.060
Pentadecanoic acid (C15:0)	0.09 ± 0.010
Palmitic acid (C16:0)	19.92 ± 0.307
Heptadecanoic acid (C17:0)	0.16 ± 0.010
Stearic acid (C18:0)	5.42 ± 0.029
Arachidic acid (C20:0)	0.13 ± 0.012
Myristoleic acid (C14:1)	0.10 ± 0.006
Palmitoleic acid (C16:1)	3.83 ± 0.137
cis-10-Heptadecenoic acid (C17:1n10)	0.11 ± 0.010
cis-9-Oleic acid (C18:1n9)	41.07 ± 0.549
Linoleic acid (C18:2 n6c)	21.08 ± 0.180
alpha-Linoleic acid (ALA, C18:3n3)	0.96 ± 0.049
gamma-Linoleic acid (C18:3n6)	0.18 ± 0.006
cis-11-Eicosenoic acid (C20:1n9)	0.41 ± 0.026
cis-11,14-Eicosadienoic acid (C20:2)	0.27 ± 0.035
cis-8,11,14-Eicosatrienoic acid (C20:3n6)	0.27 ± 0.035
Arachidonic acid (C20:4n6)	0.82 ± 0.092
Docosahexaenoic acid (DHA, C22:6n3)	0.22 ± 0.015
Others Total saturated fatty acids Total unsaturated fatty acids	4.27 ± 0.234 26.40 ± 0.40 69.33 ± 0.642

**Table 2 foods-12-00791-t002:** Effect of crocodile oil on food intake, energy intake, and body weight gain.

Parameter	Group		
	NS	NC1	NC3
Food intake (g/rat/day)	19.00 ± 0.02	18.49 ± 0.21 **	16.25 ± 0.06 **
Energy intake (kcal/rat/day)	63.34 ± 0.73	65.94 ± 1.36	66.13 ± 0.92
Body weight gain (g)	35.50 ± 2.81	41.14 ± 5.54	49.67 ± 12.20

Values are the mean ± SEM. ** *p* < 0.01 versus NS group.

**Table 3 foods-12-00791-t003:** Effect of crocodile oil on serum lipid profiles.

Parameter	Group		
	NS	NC1	NC3
Cholesterol (mg/dL)	76.13 ± 2.28	73.58 ± 7.63	69.43 ± 5.92
Triglycerides (mg/dL)	189.10 ± 33.86	100.58 ± 11.13 *	98.42 ± 6.87 **
HDL-C (mg/dL)	39.05 ± 2.70	42.74 ± 6.40	38.02 ± 3.86
LDL-C (mg/dL)	10.83 ± 1.98	13.02 ± 0.81	12.70 ± 2.73

Values are the mean ± SEM. HDL-C, high-density lipoprotein-cholesterol; LDL-C, low-density lipoprotein cholesterol. * *p* < 0.05 versus NS group. ** *p* < 0.01 versus NS group.

**Table 4 foods-12-00791-t004:** Protein identification and functional classification of unique proteins expressed in NC1 group, NC3 group, and shared proteins expressed in NC1 and NC3 groups involved with antioxidant activity.

Group	Accession Number	Gene Name	Protein Name	Function	Peptide
NC1	A0A0G2JSH9	Prdx2	Peroxiredoxin-2	Antioxidant activity; Thioredoxin-peroxidase activity	QITVNDLPVGR
NC3	P04762	Cat Cas1	Catalase	Antioxidant activity; Catalase activity; Oxidoreductase activity	GKANL
	P01946	Hba1 Hba-a1	Hemoglobin subunit alpha-1/2	Oxygen binding; Oxygen-carrier activity	LRVDPVNFK
Shared in NC1–NC3 group	Q66HF1	Ndufs1	NADH-ubiquinone oxidoreductase 75-kDa subunit	Electron-transfer activity; NADH-dehydrogenase activity	KPMVVLGSSALQR
	Q6AYA7	Rfk	Riboflavin kinase	ATP binding; Riboflavin-kinase activity;	GFGRGSK
	A0A0G2JWB6	Pxdn	Peroxidase	Peroxidase activity	EIQPGAFR

**Table 5 foods-12-00791-t005:** Protein identification and functional classification of unique proteins expressed in NC1 group involved with cognitive function.

Accession Number	Gene Name	Protein Name	Function	Peptide
Q63226	Grid2	Glutamate receptor ionotropic, delta-2	Glutamate-receptor activity; Ionotropic glutamate-receptor activity; Ligand-gated ion-channel activity	LENNMR
Q91Z79	Ppfia3	Liprin-alpha-3	Neurotransmitter secretion; Regulation of short-term neuronal; synaptic plasticity; Exocytosis of synaptic vesicles	MNDDHNK
A0A0G2K079	Bcas1	Breast carcinoma-amplified sequence 1 homolog	Myelination	DPEDTK

## Data Availability

The data presented in this study are available on request from the corresponding author.

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
