# Peer review of "Antioxidant Activity of Crocodile Oil (*Crocodylus siamensis*) on Cognitive Function in Rats"

_foods, 2023, doi:10.3390/foods12040791_

Round 1

Reviewer 1 Report

1.       Abstract should be fine-tune, the last three lines should be clear

Introduction:

1.       Introduction should start from Crocodile oil instead of ROS and antioxidants

2.       Introduction should give about where this CO production is more.

3.       What is the fatty acid composition of the CO?

4.       What is the Production and current availability and status of the CO?

5.       What are the current usage practices of the CO?

6.       Try to give a gaps

7.       Try to give the previous reports are present

8.       What is the significance of your study?

Materials and Methods:

1.       In the material check first two lines, it mentioned CO obtained and then written as “. CO was extracted using the wet cold-pressed method”

2.       Mentioned the citations in section 2.2

3.       Refer the equation 1 in the text

4.       Section 2.4, What are animals? It is about experimental animals? Or animal studies?

5.       Section 2.5, give the attributes of the rats clearly

6.       In section 2.5, How decided the dosage of CO?

7.       In section 2.6 use the formula editor for the formula and give the number of equation

8.       In section 2.6, give the citations

9.       Line 158, what is the principle of HDL and LDL automatic analyzer?

10.   Section 2.9, give appropriate citations

11.   Section 2.10.1 give appropriate citations

Results:

1.       The results are presented well

2.       Mention the Total saturated and unsaturated fats here (Table 1)

Discussion:

1.       The discussion is missing on the fatty acid composition of CO with other studies

2.       Discussion on the Free radical scavenging activity should be very deep by comparing with the other similar findings and their influence on the health

3.       Try to give the specific reasons for how CO influenced the “lipid profiles in blood”.

4.       Discuss the learning and memory abilities of rats with past studies

Conclusions:

1.       Need to re-write conclusions, try to mention each analyzed parameter-related issues

General Comments:

1.       Need to check once the Fine tuning of the tables and images

2.       Need to check the side headings

3.       Need to check the English language

Reviewer 2 Report

The manuscript entitled: “Effect of crocodile oil (Crocodylus siamensis) supplement on antioxidant properties and cognitive function in rats”, yet interesting, seems not fully fit and comply with the scope of the Journal Section (namely Nutrition) and to the special issue where it has been submitted.

Overall the connection between the antioxidant properties and impact of the studied crocodile oil on cognitive function should be better substantiated and cleared in the text. 

There are many experimental data reported, which is interesting, nonetheless the end points should be better assessed as well as the experimental design and the limits of the proposed study, e.g. the numbers of animals used for the experiments is consistent with the proposal to use the minimum number of animals, nonetheless, is this number enough to represent completely and assess the reported results?

Information on the Authority with reference to etical issues on the use of animals in the experiments should be better described. The complete diet administered to the animals should be mentioned since crocodile oil is named a supplement but other componenents could interphere with the effect of the supplement. In paragraph 2.2 a gas chromatogram could be useful to visualize the fatty acids identified as reported in the Table 1. Statistical analysis and significativity should be better detailed: please comment on this point.

Paragraph 2.3 should be better assessed. The 2,2-diphenyl-1-picrylhydrazyl test should be supported also by other tests to assess the results better. It is mentioned in the same paragraph also BHT: please explain why it is necessary as reference.

The proteomic analysis is necessary to substantiate the results: please comment on this point.

The Conclusions section for example refers to the effect of crocodile oil on triglycerides level, free radical-scavenging ability, etc. The correlation with cognitive function should be better substantiated in the text. The title of the manuscript should be rephrased to make clear which is the end point which the Authors want to reach and demonstrate, in particular please explain the correlation between the antioxidant properties and cognitive function as per the manuscript title.

Moreover, the correlation between nutrition and the observed results should be better substantiated in the text. The Conclusion section should be more detailed and report also the perspectve view of the Aytors on the area of interest and possible application. Finally, too dated References should be avoided unless necessary and justified in favor of more recent literature available data. The English language would benefit a check for better clarity and readabiliy.
